# UniTabE: A Universal Pretraining Protocol for Tabular Foundation Model in Data Science

**Yazheng Yang[†], YuQi Wang[†], Guang Liu[‡], Ledell Wu[∥⊢], Qi Liu[†]**
The University of Hong Kong[†], Beijing Academy of Artificial Intelligence[‡], Creatify AI[∥⊢]
Pokfulam Road, Hong Kong, China[†], Zhongguancun East Road, Beijing, China[‡], California, United States[∥⊢]
{yangyazh,wangyuqi}@connect.hku.hk, liuguang@baai.ac.cn,
ledell@creatify.ai, liuqi@cs.hku.hk

## ABSTRACT

Recent advancements in Natural Language Processing (NLP) have witnessed the ground-breaking impact of pretrained models, yielding impressive outcomes across various tasks. This study seeks to extend the power of pretraining methodologies to facilitating the prediction over tables in data science, a domain traditionally overlooked, yet inherently challenging due to the plethora of table schemas intrinsic to different tasks. The primary research questions underpinning this work revolve around the establishment of a universal pretraining protocol for tables with varied structures, the generalizability and transferability of learned knowledge across tasks, the adaptation to diverse downstream applications, and the incorporation of incremental columns over time. In response to these challenges, we introduce UniTabE, a straightforward yet effective method designed to process tables in a uniform manner, devoid of constraints imposed by specific table structures. UniTabE's core concept relies on representing each basic table element with a module, termed TabUnit. This is subsequently followed by a Transformer encoder to refine the representation. Moreover, our model is designed to facilitate pretraining and finetuning through the utilization of free-form prompts. In order to implement the pretraining phase, we curated an expansive tabular dataset comprising approximately 13 billion samples, meticulously gathered from the Kaggle platform. This research primarily centers on classification and regression tasks involving tabular data, and conducts rigorous experimental testing and analyses to validate the effectiveness of our methodology. The experimental results demonstrate UniTabE's superior performance against several baseline models across a multitude of benchmark datasets. This, therefore, underscores UniTabE's potential to significantly enhance the semantic representation of tabular data, thereby marking a significant stride for tabular data analysis.

## 1 INTRODUCTION

Tabular data, characterized by its structured arrangement of rows and columns, represents a foundational element within the field of data science, finding extensive practical utility in diverse domains. It facilitates a spectrum of real-world applications, such as stock market prediction, real estate price forecasting, and credit level estimation, etc. In the context of data analysis, modeling, and decision-making, classification and regression, the predominant tasks of predicting over tabular data, play a central role across diverse industries, attracting considerable attention from the research community.

However, several significant challenges have been identified within this domain: 1) The prevailing focus in the literature centers on bolstering the capabilities of powerful model architectures, often at the expense of relatively simple approaches to feature processing, such as adopting textualization [Song et al., 2019; Herzig et al., 2020; Yin et al., 2020a; Huang et al., 2020; Gorishniy et al., 2021]. This approach, while effective in certain contexts, tends to overlook the intrinsic nature and numerical significance of tabular data, particularly numerical values. This omission can impede the model's capacity to extract meaningful insights from such data, highlighting a potential gap in current methodologies [Eisenschlos et al., 2021; Herzig et al., 2020]. 2) Recent approaches have turned to finetuning large language models (LLMs) pretrained on natural language processing (NLP) datasets, such as TaBERT [Yin et al., 2020a], Tapas [Herzig et al., 2020], etc. However, LLMs do not inherently excel in handling textualized tabular data, which fundamentally differs from natural

language texts. Furthermore, these methods often employ simplistic textualization techniques mentioned above that can limit their effectiveness. 3) Prior work exploring pretraining solely on large-scale tabular datasets and assessing transferability is limited. Existing attempts [Wang and Sun, 2022] have primarily focused on a relatively small number of datasets from the same domain, which may not sufficiently validate the model's adaptability across diverse contexts. 4) Existing neural network methods often fall short in performance compared to eXtreme Gradient Boosting (XGBoost) [Chen and Guestrin, 2016] in data science applications, where XGBoost's high predictive accuracy, efficiency, and flexibility have contributed to its widespread adoption in industry. 5) Many existing approaches require strict consistency in table structures between training and testing data, posing challenges when minor modifications to table structures are necessary. This limitation becomes particularly relevant when applying trained models to tables with additional columns, a common scenario in practical applications. For example, in clinical trials, incremental columns are collected across different phases.

To address these challenges, we introduce *UniTabE*, a straightforward yet efficient framework designed to process tables uniformly while accommodating flexible table structures. UniTabE adopts a fine-grained approach to feature engineering by processing each cell in the table individually. Motivated by the success of pretraining in NLP, we explore large-scale pretraining over tables in data science, assessing the knowledge transferability and application scalability. We have collected an expansive tabular dataset from Kaggle, comprising approximately 13 billion examples spanning diverse domains. This dataset enables our large-scale pretraining over tabular data. In order to accommodate diverse pretraining and finetuning tasks within a unified framework, we introduce a universal training protocol. This protocol involves the utilization of an auto-regressive decoder, coupled with adaptable free-form prompts. The decoder performs reasoning on the high-level semantic representation produced by the encoder, functioning as an adaptable module capable of task-specific customization through the application of task-specific prompts. To ensure the preservation of a significant portion of pretrained knowledge within the encoder that serves as the foundational module designed for tabular data, we employ a shallow decoder. This deliberate choice compels the encoder to assume the primary responsibility for absorbing the intricacies of learning during pretraining. To comprehensively evaluate the effectiveness of our method, we conduct extensive experiments under different scenarios including predictive tasks on prominent benchmark datasets, filling in missing values, zero-shot prediction, adaptation to incremental tables, and integration of the neural semantic representation with XGBoost, etc.

Our contributions are as follow:

- We introduce an innovative architectural framework, UniTabE, specifically designed to offer meticulous feature processing tailored to tabular data. Furthermore, through the incorporation of free-form prompts into our model, we augment its scalability to an extensive array of tasks for downstream applications.

- We have built an extensive tabular dataset for large-scale pretraining. We introduce an efficient framework for both pretraining and finetuning, optimized to leverage the full potential of our collected dataset.

- With comprehensive experiments, we substantiate the feasibility of pretraining on tabular data, underscore the transferability of knowledge acquired, and highlight the substantial performance enhancements it affords in downstream tasks. Our experimental findings also elucidate the practical efficacy of our approach in scenarios such as missing value handling, zero-shot prediction, and adaptability to incremental column structures. Moreover, our method outperforms XGBoost across a wide spectrum of benchmark datasets, demonstrating its superiority.

In Sec. 2 we cover related work. Sec. 3 introduces the model architecture of our proposed UniTabE. Sec. 4 gives detailed description of our pretraining and finetuning strategies. Finally, Sec. 5 provides the results of experimental studies which verify the effectiveness of our method.

## 2 RELATED WORK

### 2.1 TABLE DATA FEATURIZATION

Tables encompass various data types, and to process this tabular data with neural models, conventional methods typically convert each data format into a continuous space using individual strategies. For instance, texts are processed with tokenization and word embedding, while images are handled using image patch

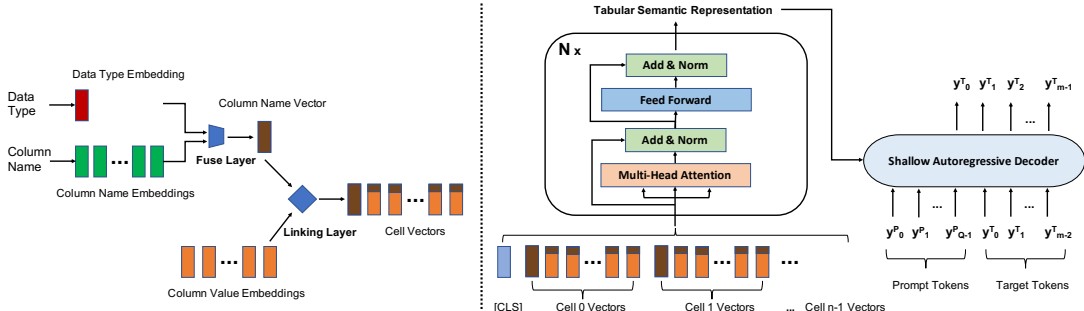

Figure 1: The left part delineates the operational procedure of TabUnit module in processing individual cells. The right part provides an overview of our UniTabE architecture. "n" denotes the number of cells in each example, "Q" denotes the length of prompt, while "T" here represents the length of target. A shallow decoder is applied offering adaptability to a spectrum of diverse downstream tasks.

embedding. However, prior research tends to simplify this process by treating numerical values as text and then applying the same embedding strategy, which invariably disrupts the original recording structure and numerical meanings [Eisenschlos et al., 2021; Herzig et al., 2020]. Given the prevalence of numerical values in tables, this area has garnered increasing attention [Gorishniy et al., 2022]. To enhance the representation of numerical values, MATE [Eisenschlos et al., 2021] and TaPas [Herzig et al., 2020] introduced a ranking embedding based on numeric rank, which relies on comparisons. Further, TUTA [Wang et al., 2021] applied additional numerical features, such as magnitude, precision, the first digit, and the last digit, to distinguish numbers from text. Gorishniy et al. [2022] attempted to train the embedding vector for numbers. Wang and Sun [2022] suggested categorizing tabular data into three distinct types: textual, categorical, and numerical. Their model, TransTab, concatenates columns of the same type into a text sequence, with column names, column values, and different columns separated by a space. After the concatenation, these three text sequences are fed into the embedding layer individually.

## 2.2 PRETRAINING TABLE MODELS

In recent years, the pretraining of language models (LMs) over vast text corpora has led to noteworthy enhancements in performance for a variety of downstream tasks. This success has stimulated a growing body of work that focuses on pretraining and adapting LMs specifically for tabular data. The prevailing method employed by these studies is to fine-tune LMs that have been pretrained on NLP datasets, such as BERT [Devlin et al., 2018], BART [Lewis et al., 2019], etc. Typically, this training utilizes the Masked Language Model (MLM) objective, as evidenced by models like Tabtransformer [Huang et al., 2020], TaBERT [Yin et al., 2020b], Tabnet [Arik and Pfister, 2021], Saint [Somepalli et al., 2021], and so on. Liu et al. [2022] proposed a modality transformation that converts tabular data into textual data using a basic lexicon and ordered syntax before feeding it into the pretrained LMs. They then finetuned their model using the same MLM objective. However, LMs pretrained on natural texts do not perform optimally as the textualized tabular data differs fundamentally from natural language texts. The efficacy of finetuned LMs without architectural modifications has largely been confined to text data so far. But making modifications might lead to undesired outcomes like catastrophic forgetting of knowledge learned from natural language corpora [Chen et al., 2020; Bavarian et al., 2022]. In this work, our focus lies on pretraining a large model from scratch on tabular data. In line with previous work, we construct our model upon the renowned Transformer encoder, utilizing a self-supervised objective similar to MLM, which masks parts of the model input and then predicts the masked content. Contrary to previous work, we abstain from textualizing the tabular data using a simplistic strategy. Instead, we introduce a TabUnit module designed to process the basic element of a table independently, leading to an improved modeling of tabular data.

## 3 UNITABE ARCHITECTURE

In this section, we present the architecture of our model, which is composed of three primary components: the *TabUnit*, the *Encoding Layer*, and a *Shallow Decoder*. Our model employs the TabUnit module serving as the foundational feature processor for varying data types before leveraging the Transformer's encoder for

further encoding. Through adopting the setting of prompts and integrating a decoder, our model becomes adaptable to a wide range of tasks.

**TabUnit Module** As depicted on the left side of Figure 1, we propose the use of an unified module, called *TabUnit* in this work, for modeling the basic element of tabular data. To mitigate the influence of table structure, we treat each cell in a table as a key-value pair representing the column name and the column value respectively. In our implementation, we concatenate the vector of column name and vectors of cell value into a sequence as the cell's representation. To get the column name vector, the tokens of the column name $X_{cn}$ are first passed into the embedding module:

$$\mathbf{x}_{cn} = Avg(Emb(X_{cn}))$$
$$\mathbf{x}_{dt} = Avg(Emb_{DT}(X_{dt})) \tag{1}$$

where $Emb$ represents the embeddings consisting of word embedding and positional embedding. $\mathbf{x}_{cn}$ is the vector after mean pooling across the dimension of token sequence. We map data types $X_{dt}$ into integers (e.g. numerical→0, categorical→1, and textual→2) and subsequently employ an embedding module $Emb_{DT}$ to obtain the data type embedding vector $\mathbf{x}_{dt}$. Here, the $Emb_{DT}$ is specifically designed to help the model adeptly handle diverse data formats, particularly in instances where columns share the same/similar name but contain values in different data formats. For example, the values in the "salary" column of a table in a downstream task might be numerical, while those in the corresponding column of another table could be textual (e.g., high income, medium income, and low income, etc). In some way, such situation may cause confusion for the neural networks. In order to remedy this problem, our model integrates the data type information into $\mathbf{x}_{cn}$ via a fuse layer relied on the gate mechanism:

$$g_{dt} = \text{Sigmoid}(\mathbf{v}_{fl}\text{ReLU}(\mathbf{w}_{fl}^{\top}\mathbf{x}_{dt} + \mathbf{b}_{fl}))$$
$$\mathbf{v}_{cn} = f_{fl}(\mathbf{x}_{cn}, \mathbf{x}_{dt}) = (1 - g_{dt})\mathbf{x}_{cn} + g_{dt} * \mathbf{x}_{dt} \tag{2}$$

where $\mathbf{w}_{fl}$, $\mathbf{b}_{fl}$ and $\mathbf{v}_{fl}$ are trainable parameters. Note that the ratio $g_{dt}$ is calculated only conditioning on $\mathbf{x}_{dt}$. Theoretically, integrating an equal amount of data type information into the column name representation across columns of the same data type is reasonable, as opposed to computing the fusing ratio $g_{dt}$ based on both $\mathbf{x}_{cn}$ and $\mathbf{x}_{dt}$ that will lead to adding varied amount of data type information. Tokens in each cell are also passed into the embedding module:

$$\{\mathbf{x}_{cv}^0, \mathbf{x}_{cv}^1, \mathbf{x}_{cv}^2, ..., \mathbf{x}_{cv}^{q-1}\} = Emb(\{x_{cv}^0, x_{cv}^1, x_{cv}^2, ..., x_{cv}^{q-1}\}) \tag{3}$$

Where the embedding module $Emb(.)$ is shared to carry out embedding for column names and column values, $q$ here denotes the length of the cell value. Given the orderless nature of self-attention within the Transformer encoder, it becomes challenging for the model to learn the connection between the column name and its value when all cells are concatenated into a sequence. As a result, we introduce a *Linking Layer* to establish the relationship within each (column name, cell value) pair. We employ a gated function to weave the information from the column name into its corresponding value. Thus guaranteeing vectors coming from the same pair have higher weights in the self-attention operation to make the model aware of the association between column name and values.

$$\alpha = \text{Sigmoid}(\mathbf{v}_{lk}\text{ReLU}(\mathbf{w}_{lk}^{\top}\mathbf{v}_{cn} + \mathbf{b}_{lk}))$$
$$\mathbf{v}_{cv}^i = \mathbf{x}_{cv}^i + \alpha * \mathbf{v}_{cn} \tag{4}$$

where $\mathbf{w}_{lk}$, $\mathbf{b}_{lk}$ and $\mathbf{v}_{lk}$ are learnable parameters. We employ $\alpha$ to ensure that an equal amount of column name information is integrated into all value vectors. This allows the model to recognize which parts of vectors are values corresponding to specific column. To avoid having the value information overshadowed by the column name information, we only apply the multiplication operation to $\alpha$ and $\mathbf{v}_{cn}$. Overall, the TabUnit can be briefly formulated as:

$$\mathbf{X}_{TU} = \{\mathbf{v}_{cn}, \mathbf{v}_{cv}^0, \mathbf{v}_{cv}^1, ..., \mathbf{v}_{cv}^{q-1}\} = f_{TabUnit}(X_{dt}, X_{cn}, X_{cv}) \tag{5}$$

where $X_{dt}$, $X_{cn}$ and $X_{cv}$ denote the data type indicator, tokens of column name and tokens of column value, respectively. The concatenation of column name vector and value vectors is treated as the inner representation of a tabular cell. In our implementation, all cells are processed in parallel.

**Encoding Layer** We concatenate the representations of all cells, and attach a trainable [CLS] vector to the head of such sequence. We leverage the Transformer encoder as the encoding layer:

$$\{\mathbf{h}_{cls}, \mathbf{h}^0, \mathbf{h}^1, ..., \mathbf{h}^{N-1}\} = f_{Enc}(\mathbf{v}_{cls}, \mathbf{X}_{TU}^0, \mathbf{X}_{TU}^1, ..., \mathbf{X}_{TU}^{n-1}) \tag{6}$$

where $n$ is the number of cells, and $N$ is the length after concatenating all cells' representations.

**Shallow Decoder** During pretraining, we want to encourage the encoder to store most of the learned knowledge. Hence, we adopt a Long Short-Term Memory network (LSTM) [Hochreiter and Schmidhuber, 1997] as the weak decoder. Specifically, the hidden state of [CLS] token and the prompt are passed to the decoder to compute the initial state of our decoder:

$$\{\mathbf{y}_0^P, \mathbf{y}_1^P, ..., \mathbf{y}_{Q-1}^P\} = f_{attn}(\mathbf{W}_1^\top Emb(\{y_0^P, y_1^P, ..., y_{Q-1}^P\}), \mathbf{W}_2^\top \{\mathbf{X}_{TU}^0, \mathbf{X}_{TU}^1, ..., \mathbf{X}_{TU}^{n-1}\}) \tag{7}$$

$$\mathbf{v}_p = \sum_i \frac{\exp(\mathbf{v}_1^\top \mathbf{y}_i^P)}{\sum_j \exp(\mathbf{v}_1^\top \mathbf{y}_j^P)} \mathbf{y}_i^P \tag{8}$$

$$\mathbf{v}_{state} = \mathbf{W}_s^\top \{\mathbf{v}_p, \mathbf{h}_{cls}\} + \mathbf{b}_s \tag{9}$$

where the $f_{attn}$ denotes the dot product attention [Vaswani et al., 2017]. $\mathbf{W}_1, \mathbf{W}_2, \mathbf{v}_1, \mathbf{W}_s$ and $\mathbf{b}_s$ are trainable parameters. Here $\mathbf{v}_p$ represents the weighted average of attention states of the prompt. The embedding layer of decoder also share the same parameter as that one on TabUnit to further reduce parameters of the decoder. The target sequence of tokens are generated by the decoder step by step conditioned on the initial state and previously produced token.

## 4 PRETRAINING & FINETUNING

In this section, we will commence by presenting the pretraining objectives that consist of multi-cell-masking and contrastive learning, followed by an elaboration on the finetuning methodologies.

### 4.1 PRETRAINING OBJECTIVE

**Multi-Cell-Masking.** Previous research in NLP pretraining utilized self-supervised tasks within datasets to provide pretraining signals, such as predicting next token, generating masked spans of texts, and determining the subsequent sentence, etc. These studies have shown that unlabeled data can aid in the learning of semantic representation. In this work, we also adopt the mask-then-predict approach to facilitate self-supervised training. In practical applications, filling in the entire content of a cell is more useful than merely filling in part of the cell content. As such, we treat each cell as the basic masked unit, as opposed to the token level in NLP pretraining. We randomly replace the content of the masked cells with the special token, [MASK]. We also use [MASK] as the default content for cells whose values are missing.

For each example, we arbitrarily mask columns and train the model to predict the masked content. Often, there may be multiple missing values in downstream tasks. Therefore, we also train the model under conditions where several values are missing, to familiarize the model with such situations. The number of masked cells varies during training. The prompt template for pretraining is set to "fill in missing value, <column name> :", which specifies the precise masked column to predict. The details regarding the number of randomly masked cells and their corresponding probabilities will be elaborated in § 5.2. An illustration of the multi-cell-masking objective is presented in Figure 5. Our model is trained with the optimization objective of the maximum log likelihood estimation.

**Contrastive Learning.** Expect for predicting values of masked cells that focuses on the global information of the table, we can also concentrate on local properties. In various applications, table row represents a sample and its columns are features. A natural self-supervised optimization signal would be distinguishing the differences across rows. Prior work [Khosla et al., 2020; Tian et al., 2020] in computer vision treated the augment of original image as the positive sample, and augments of other images as negative ones. For instance, the augment can be a block of the original image. Inspired by prior work, we

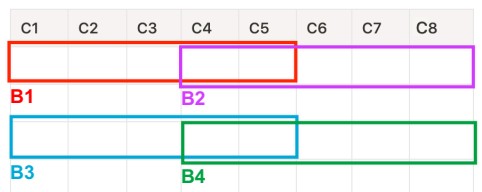

Figure 2: Demonstration of contrastive learning. (B1, B2) is positive pair, while (B1, B3) and (B1, B4) are negative pairs.

regard a subset of row as the compared unit. In Figure 2, block B2 is positive example, B3 and B4 are negative examples when B1 is treated as an anchor. B3 and B4 come from different row that is not identical to the row containing the given anchor. We utilize the cosine similarity to measure the closeness between two blocks. The training objective is maximizing the similarity between anchor and positive block while

Table 1: Statistics of our pretraining dataset. "**Avg# NC**", "**Avg# CC**" and "**Avg# TC**" signify the average counts of numerical columns, categorical columns, and textual columns, respectively, within each table. The lower section of the table exhibits the Top-5 domains most prominently featured in our dataset.

| Domains | # Domains | # Tables | # Examples | Avg# NC | Avg# CC | Avg# TC |
|---|---|---|---|---|---|---|
| ALL | 303 | 283K | 13B | 28.7 | 0.4 | 7.7 |
| Investing | 1 | 71K | 1B | 29.33 | 0.02 | 1.58 |
| Time Series | 1 | 65K | 1B | 6.47 | 0.02 | 2.27 |
| Finance | 1 | 52K | 773M | 37.57 | 0.04 | 1.46 |
| Economics | 1 | 47K | 488M | 40.34 | 0.01 | 1.27 |
| Games | 1 | 32K | 430M | 23.37 | 0.66 | 3.66 |

minimizing it between anchor and negative block. We adopt the overlap between B1 and B2 to guarantee such supervised signal will not make the optimization collapse contributing to the robustness for various tables.

## 4.2 FINETUNING FORMULATION

**Filling in Missing Value as Prediction.** When finetuning our model for downstream tasks, we can consider the target as an additional column of the table. The model is then tasked with predicting the masked values of this target column using the same prompt as during pretraining. Thanks to the decoder, UniTabE is capable of generating textual and numerical targets. For example, the trained model is suited for classification and regression tasks, as well as predicting missing values in tables. In our implementation, we also support constrained generation. This feature is particularly beneficial for classification tasks, as the model only needs to predict from a small subset of the vocabulary.

**Finetuning with Task-specific Prompt.** Apart from those tasks where we treat the target as the masked column of the table, there are tasks that require the model to perform reasoning over the table and other inputs. For instance, in table question answering (TableQA) tasks, the model needs to produce an answer conditioned on the provided table and question. The prompt used in these tasks may include the task instruction and the question, appropriately formatted.

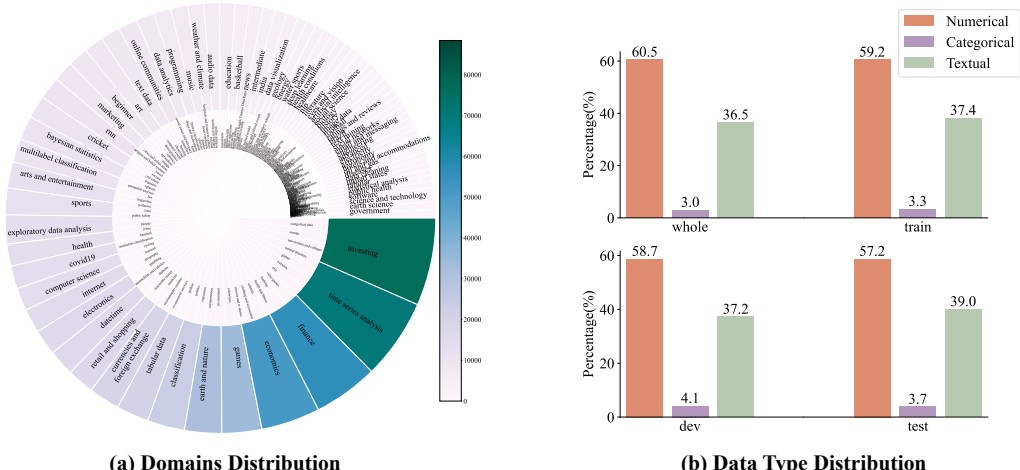

**(a) Domains Distribution**        **(b) Data Type Distribution**

Figure 3: Distribution visualization. The left part (a) demonstrates the distribution of domains and the number of tables in each domain. Please magnify the figure as some captions are small. The right part shows the proportion (cell level) of different data types in train/dev/test splits.

## 5 EXPERIMENTS AND ANALYSES

In this section, we will commence by introducing the pretraining dataset collected from Kaggle. Subsequently, we will provide a detailed exposition of our implementation. Finally, we will engage in an extensive discussion of the experimental results and analyses, encompassing an assessment of performance in Kaggle benchmarks, public benchmarks, zero-shot prediction, and related aspects.

Table 2: Performance comparison in 12 Kaggle tasks coming from different domains. "UniTabE scratch" is trained from scratch without pretraining. The result of each method is the average score across five runs.

| Method/Dataset | Classification (AUC ↑) | | | | | | Regression (R2 ↑) | | | | | |
|---|---|---|---|---|---|---|---|---|---|---|---|---|
| | PID | RWQ | HFP | HIC | EPL | LDP | MIP | GPP | RSP | CCL | MMP | HPA |
| XGBoost | 0.79 | **0.67** | **0.89** | 0.85 | 0.73 | 0.50 | 0.54 | 0.64 | 0.98 | 0.78 | 0.71 | 0.49 |
| TransTab-LSTM | 0.81 | 0.56 | 0.74 | 0.85 | 0.73 | 0.50 | 0.50 | 0.55 | 0.64 | 0.65 | 0.51 | 0.47 |
| UniTabE scratch | 0.76 | 0.57 | 0.61 | 0.85 | 0.73 | 0.52 | 0.53 | 0.76 | **0.99** | 0.72 | 0.82 | 0.54 |
| UniTabE finetune | **0.83** | 0.66 | 0.81 | **0.86** | **0.78** | **0.53** | **0.75** | **0.99** | **0.99** | **0.96** | **0.87** | **0.58** |

Table 3: Evaluation results with AUC on public tabular datasets. We conducted five training iterations for each method and subsequently computed the average scores across these runs.

| Method/Dataset | CG | CA | DS | AD | CB | BL | IO | Avg. |
|---|---|---|---|---|---|---|---|---|
| XGBoost | 0.78 | 0.93 | 0.54 | **0.91** | 0.87 | 0.82 | 0.71 | 0.79 |
| NODE [Popov et al., 2019] | 0.65 | 0.85 | 0.58 | 0.90 | 0.70 | 0.83 | 0.67 | 0.74 |
| AutoInt [Song et al., 2019] | 0.77 | 0.93 | 0.57 | **0.91** | 0.82 | **0.84** | 0.72 | 0.79 |
| Tapas [Herzig et al., 2020] | 0.77 | 0.92 | 0.63 | **0.91** | 0.87 | **0.84** | 0.70 | 0.81 |
| TaBERT [Yin et al., 2020a] | 0.72 | 0.88 | 0.54 | 0.90 | 0.71 | 0.82 | 0.66 | 0.75 |
| TabTransformer [Huang et al., 2020] | 0.76 | 0.92 | 0.56 | **0.91** | 0.82 | 0.82 | 0.71 | 0.79 |
| FT-Transformer [Gorishniy et al., 2021] | 0.77 | 0.93 | 0.58 | **0.91** | 0.85 | **0.84** | 0.71 | 0.80 |
| TabNet [Arik and Pfister, 2021] | 0.49 | 0.48 | 0.48 | 0.90 | 0.53 | 0.79 | 0.62 | 0.61 |
| TUTA [Wang et al., 2021] | 0.74 | 0.92 | 0.61 | 0.81 | 0.74 | 0.83 | 0.65 | 0.76 |
| TransTab [Wang and Sun, 2022] | 0.73 | 0.86 | 0.52 | 0.90 | 0.80 | 0.71 | 0.73 | 0.75 |
| TransTab-LSTM | 0.70 | 0.85 | 0.56 | 0.90 | 0.72 | 0.83 | 0.73 | 0.76 |
| TabPFN [Hollmann et al., 2022] | **0.79** | 0.93 | 0.62 | 0.88 | 0.78 | 0.81 | 0.70 | 0.79 |
| GANDALF [Joseph and Raj, 2023] | 0.78 | 0.92 | 0.63 | **0.91** | 0.84 | **0.84** | 0.71 | 0.80 |
| Llama2 13B [Touvron et al., 2023] | 0.77 | 0.92 | 0.65 | **0.91** | 0.84 | 0.83 | 0.68 | 0.80 |
| UniTabE scratch | 0.76 | 0.93 | 0.62 | **0.91** | 0.85 | **0.84** | 0.74 | 0.81 |
| UniTabE + XGBoost | **0.79** | 0.93 | 0.60 | **0.91** | **0.88** | 0.83 | 0.74 | 0.81 |
| UniTabE finetune | **0.79** | **0.94** | **0.66** | **0.91** | **0.88** | **0.84** | **0.76** | **0.83** |

## 5.1 PRETRAINING DATASET

As there are no large-scale, high-quality tabular datasets available for pretraining, we have collected our own dataset by crawling from Kaggle. We downloaded CSV tables, omitting empty columns. Specifically, we constructed initial keywords for each domain and extended the set of keywords using WordNet under the same topic. We then searched and downloaded tables using these keywords. For tables originating from the same Kaggle dataset, we attempted to join tables using primary and foreign keys. This process resulted in a 7TB dataset containing 13 billion tabular examples. Statistics about the pretraining data are presented in Table 1. We also present the statistics of the Top-5 domains in this table. Figure 3 shows the distribution of domains and the proportion of cells of different data types for each split of our dataset. We tried to split our dataset in such a way as to maintain a similar distribution of data types and domains.

## 5.2 IMPLEMENTATION DETAILS

We train our UniTabE with 32 A100 GPUs in the distributed way. Our model is implemented with the PyTorch v1.12. The Transformer encoder used as the backbone of our model is borrowed from the huggingface "transformers" module. During training, the learning rate is set to be 1e-5, the batch size is 64. We use Adam [Kingma and Ba, 2015] as the optimizer with $\beta_1$=0.9, $\beta_2$=0.999 and $\epsilon$ =$10^{-8}$. We train three variants of our model, $UniTabE_{base}$, $UniTabE_{large}$ and $UniTabE_{xlarge}$. The hidden size and embedding size for $UniTabE_{base}$ are both set as 768. Its encoder is the 12 layers of self-attention stack with 12 heads. $UniTabE_{large}$ consists of 24 layers encoder, 16 attention heads. Its hidden size and embedding size are both 1024. $UniTabE_{xlarge}$ is the 48 layers' version of $UniTabE_{large}$. More details about implementation of baselines and experimental setting, please refer to Appendix A.3.

## 5.3 RESULTS & ANALYSES

Many practical tabular tasks in the field of data science fall into the categories of classification or regression. As such, we evaluate the effectiveness of our pretrained model on these types of tasks. Details about baselines are attached in the Appendix A.1.

**Kaggle Benchmarks.** We selected a diverse set of tasks comprising 6 representative classification assignments and 6 regression tasks from Kaggle. To ensure a fair and unbiased evaluation, we deliberately excluded these specific tabular datasets from our pretraining corpus. This precaution was taken to prevent the pretrained model from gaining undue familiarity with the target columns, thereby avoiding any potential bias in the results. The statistical metrics pertaining to these datasets are delineated in Table 7. The comparison of results among methods is presented in Table 2. On most of datasets, we notice that our method outperforms XGBoost that is widely used in industry due to its promising performance. TransTab-LSTM is the model that we equip TransTab with the same decoder as ours. Its vital differences from ours consist of the feature processing and gate mechanism integrated with the Transformer encoder. Compared with TransTab-LSTM, our UniTabE achieves higher scores in both classification and regression tasks. It demonstrates the superiority of our table feature processor, TabUnit, as opposed to the table textualization strategy of TransTab. This is indicative of the efficacy of our model's approach to handling and understanding the nuanced structure of tabular data.

**Public Benchmarks.** Since those 12 tasks coming from Kaggle might have the similar domains to our pretraining data, we also use other public tabular datasets (CG, CA, DS, AD, CB, BL and IO) to further evaluate the efficacy of our method. These datasets are all binary classfication tasks. The statistical data pertaining to these datasets is presented in Table 7. We compare our method against previous start-of-the-art (SOTA) methods for tabular classification on these datasets. Table 3 presents experimental results on these datasets. Similar to those results in 12 Kaggle tasks, our method also achieves impressive results indicating its superiority in learning the intrinsic semantic information of tabular data.

Table 4: The accuracy of zero-shot classification.

| Method/Dataset | CG | CA | DS | AD | CB | BL | IO |
|---|---|---|---|---|---|---|---|
| UniTabE finetune | 0.75 | 0.89 | 0.62 | 0.86 | 0.78 | 0.80 | 0.94 |
| Random Initial | 0.30 | 0.41 | 0.50 | 0.54 | 0.41 | 0.26 | 0.06 |
| Zero-Shot | **0.70** | **0.56** | **0.58** | **0.76** | **0.57** | **0.73** | **0.94** |

**Zero-shot Prediction.** Table 4 presents the results of zero-shot prediction. The "Random Initial" approach, which does not load the pretrained parameters, exhibits inferior performance. In contrast, our pretrained model demonstrates strong performance, even without fine-tuning on the target datasets, indicating its promising generalizability. These results also suggest that UniTabE acquires a certain degree of high-level reasoning capabilities through extensive large-scale pretraining, which contributes to its robust performance in zero-shot settings.

Table 5: The percentage(%) of AUC drop facing with incremental columns.

| Method/Drop k | 1 | 2 |
|---|---|---|
| TransTab-LSTM | 7.5 | 11.7 |
| UniTabE scratch | 5.3 | 8.4 |
| UniTabE finetune | **3.5** | **5.8** |

**Adaption of Incremental Columns.** We want to investigate the scalability of various models when confronted with an increasing number of columns. To simulate this scenario, we remove k columns from the original training set and train models using the modified data. Subsequently, we perform inference using the unaltered test set. We conduct a comparative analysis between our methodology and TransTab-LSTM, a representative approach that necessitates congruence in table structure between the training and testing data. The results of this experiment are displayed in Table 5. Thanks to the flexibility afforded by the TabUnit component of UniTabE, our model exhibits adaptability to the introduction of new columns with a relatively minor performance deterioration.

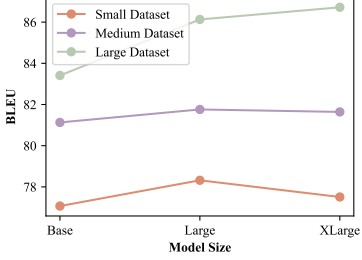

Figure 4: BLEU scores illustrating the generation of textual values across various model sizes and dataset sizes.

**Model Size Analysis.** We want to see the impact of the size of UniTabE on performance across different size of dataset. 50 tables for each dataset size are reserved and not included in the pretraining dataset. Here "small" means datasets whose example numbers do not surpass 2000, and "medium" means datasets of which the example numbers are between 2000 and 20000, while "large" denotes those datasets whose data numbers exceed 20000. The results presented in Figure 4 indicate that for larger finetuning datasets, larger models (like $\text{UniTabE}_{xlarge}$) tend to perform better. However, for smaller datasets, the performance of larger models tends to decrease. We notice that $\text{UniTabE}_{large}$ is a balance size for different

datasets. Hence, we apply the $\text{UniTabE}_{large}$ to compare the performance with baselines in subsequent experiments.

**Ablation Analysis.** Table 6 provides the results from various ablation studies. We notice that the omission of the fuse layer from the TabUnit module leads to a decline in performance, suggesting the significance of incorporating data type information into the cell representation. Similarly, removing the linking layer, responsible for associating column names with column values, makes the model oblivious to the distinction between vectors of column names and values. This is confirmed by a notable decrease in the AUC score, attesting to its efficacy. We further expel the contrastive learning and multi-cell masking objectives separately. A significant performance decline is observed when the former is removed, indicating its vital role in discerning the intrinsic differences within the table. The reduced performance following the removal of the multi-cell masking objective further emphasizes its importance in tabular pretraining. Upon evaluating models pretrained with varying masking ratios, a masking ratio of 0.15 yields an average AUC score of 0.83, proving better for the given tasks, while ratios of 0.2 and 0.3 appear inordinate. Excessive masking can impede accurate prediction of masked values and hinder differentiation between positive and negative pairs in contrastive learning. Furthermore, substituting UniTabE's 1-layer LSTM decoder with a 1-layer Transformer decoder showed the former to be more suitable for the tasks at hand. Models with an increasing number of LSTM decoder layers exhibit a declining trend in performance, suggesting a simpler decoder enables the encoder to retain more knowledge, leading to enhanced semantic representation of tabular data and superior performance. Additional exploration on the text generation capabilities of models with varying decoder layers is presented in Appendix A.5. The task involves querying the model about reasons for loan applications. The results shown in Table 9 indicate that models with additional decoder layers achieve improved BLEU scores, suggesting that more complex decoders absorb a greater portion of the pretrained knowledge, potentially detracting from the encoder. This can subsequently reduce performance in classification tasks that are reliant on the semantic representation of tabular data generated by the encoder.

Table 6: Ablation studies. The average AUC score is computed across 7 datasets(CG ∼ IO). "MCM" and "CL" denote multi-cell masking and contrastive learning objective.

| Method | Avg. |
|---|---|
| UniTabE | **0.83** |
| w/o Fuse Layer | 0.77 |
| w/o Linking Layer | 0.75 |
| w/o Fuse&Linking | 0.72 |
| w/o MCM | 0.81 |
| w/o CL | 0.79 |
| MCM rate: 0.3 | 0.81 |
| MCM rate: 0.2 | 0.81 |
| MCM rate: 0.1 | 0.82 |
| 3L Decoder | 0.81 |
| 6L Decoder | 0.79 |
| 12L Decoder | 0.78 |
| 1L Transformer Decoder | 0.81 |

**Learned Feature + XGBoost.** We are interested in examining the synergistic effects of combining the semantic representation derived from UniTabE with traditional machine learning algorithms, such as XG-Boost. We integrate the original features with the generated representation vector, and use this composite data as input to XGBoost. The outcomes are displayed in Table 3. These results indicate that the neural feature acts as a beneficial supplement, leading to a slight performance enhancement in most instances.

**Fill in Missing Value Analysis.** Furthermore, we assess the efficacy of our approach concerning its capacity to fill in missing values, with the corresponding results displayed in Table 10. The improvement in performance provides additional experimental support for the effectiveness of our pretrained model in addressing missing values, thereby affirming its potential utility in applications related to data completion, data recovery and tabular data synthesis tasks.

# 6 CONCLUSIONS

In this research, we investigate the challenge of pretraining large-scale models specifically on tabular data. We demonstrate the feasibility and scalability of such pretraining. To address the inherent difficulties in pretraining over tabular data, we introduce UniTabE, a flexible approach capable of conducting fine-grained feature processing as well as adapting to different tasks with single framework. We have also collected a substantial dataset, comprising 13B examples spanning various domains, for pretraining purposes. To ascertain the effectiveness of our methodology, we carry out extensive experiments across 19 downstream tasks, with the outcomes confirming the superiority of our approach. Additionally, we explore issues related to the practical application of our method, including zero-shot prediction, adaptation to incrementally added columns, and combination of learned representation and traditional machine learning method (XGBoost).

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

# A  APPENDIX

## A.1  BASELINES

**XGBoost** [Chen and Guestrin, 2016] [1] Despite a large number of proposed neural network architectures for tabular data, the performance gap between them and the "shallow" ensembles of decision trees, like XGBoost, often remains significant in industry. We use it as baseline for classification tasks. Implemented based on the XGBoost package.

**TransTab** [Wang and Sun, 2022] [2] also builds upon the Transformer encoder. It splits data into three categories: numerical, binary and categorical. Before being processed by the encoder, columns of the same data type are textualized into a text by inserting a space among columns, column names and column values. For instance, textual columns will be organized as the form: "column0_name column0_value column1_name column1_value ... columnk_name columnk_value". In addition, the numerical values are represented through multiplying with a training vector.

**TransTab-LSTM** In order to equip TransTab with the ability of text generation for the pretraining objective of multi-cell masking, we integrate the LSTM decoder of our model into TransTab. Thus, we compare the performance of different tabular data featuring strategies.

**Linear Regress** [3] is also used to compare the performance for regression tasks. We adopt the Scikit-learn implementation of linear regression in the experiments.

**Tapas** [Herzig et al., 2020] [4] extends the pretrained BERT to encompass the encoding of tabular data. This extension is realized through the joint pre-training process that encompasses both textual segments and tabular data extracted from Wikipedia. TAPAS is a method devised for addressing questions posed over tables, and notably, it operates without the generation of explicit logical forms. The training procedure for TAPAS is conducted under the paradigm of weak supervision, and its predictive mechanism involves the selection of specific cells within tables, optionally accompanied by the application of an associated aggregation operator to the chosen cell selections.

**TabPFN** [Hollmann et al., 2022] [5] TabPFN, denoting a Prior-Data Fitted Network (PFN), undergoes offline training as a one-time process, aimed at approximating Bayesian inference. This approximation is applied to synthetic datasets generated from a prior distribution that incorporates principles rooted in causal reasoning, encompassing a vast array of structural causal models, with an inherent preference for simplicity. It employs in-context learning (ICL), acquiring the capability to make predictions based on sequences of labeled examples within the input context, all while obviating the need for subsequent parameter updates. This model operates without necessitating hyperparameter tuning, while showcasing competitive performance against contemporary state-of-the-art classification methodologies.

**TaBERT** [Yin et al., 2020a] [6] TaBERT is a pretrained language model designed for learning joint representations of natural language utterances and structured tables. TaBERT can be integrated as a substitute for the original encoder within a semantic parser, facilitating the computation of representations for both utterances and table columns.

**Llama2 13B** [Touvron et al., 2023] [7] The recent advanced and versatile language model that offers enhanced capabilities, increased language support, and improved performance across a wide range of natural language processing tasks. This state-of-the-art model leverages the latest advancements in deep learning to provide comprehensive linguistic insights.

**TabTransformer** [Huang et al., 2020] [8] The TabTransformer architecture is based on the Transformer, with its primary objective being the acquisition of effective contextual embeddings for categorical features of tabular data. Its experiments reveal the resilience of contextual embeddings derived from the TabTransformer

---

[1] xgboost.sklearn

[2] https://github.com/RyanWangZf/transtab

[3] https://scikit-learn.org/stable/modules/classes.html#module-sklearn.linear_model

[4] https://huggingface.co/docs/transformers/model_doc/tapas

[5] https://github.com/automl/TabPFN

[6] https://github.com/facebookresearch/TaBERT

[7] https://huggingface.co/meta-llama/Llama-2-13b-hf

[8] https://github.com/lucidrains/tab-transformer-pytorch

Table 7: Statistics of datasets in downstream tasks. "**# NC**", "**# CC**", "**# TC**" and "**# Examples**" denote the number of numerical columns, categorical columns, textual columns of the table and the total number of examples in the dataset, respectively.

| Name | # NC | # CC | # TC | # Examples |
|------|------|------|------|------------|
| Pima Indians Diabetes (PID) [url] | 8 | 0 | 0 | 768 |
| Red Wine Quality (RWQ) [url] | 11 | 0 | 11 | 1599 |
| Heart Failure Prediction (HFP) [url] | 7 | 5 | 0 | 299 |
| Health Insurance Cross Sell (HIC) [url] | 7 | 0 | 3 | 381109 |
| Eligibility Prediction for Loan (EPL) [url] | 6 | 3 | 3 | 614 |
| Loan Default Prediction (LDP) [url] | 24 | 3 | 7 | 67463 |
| Medical Insurance Payout (MIP) [url] | 4 | 2 | 1 | 1338 |
| Gold Price Prediction (GPP) [url] | 80 | 0 | 1 | 1718 |
| Reliance Stock Price (RSP) [url] | 10 | 0 | 5 | 6205 |
| Credit Card Limit Prediction (CCL) [url] | 7 | 3 | 1 | 400 |
| Miami Housing Prediction (MMP) [url] | 14 | 0 | 3 | 13932 |
| House Prices - Advanced (HPA) [url] | 37 | 2 | 41 | 1460 |
| Credit-G (CG) [url] | 7 | 2 | 11 | 1000 |
| Credit-Approval (CA) [url] ) | 6 | 3 | 6 | 690 |
| Dress-Sales (DS) [url] | 1 | 0 | 11 | 500 |
| Adult (AD) [url] | 2 | 0 | 12 | 48842 |
| Cylinder-Bands (CB) [url] | 13 | 4 | 18 | 540 |
| Blastchar (BL) [url] | 3 | 5 | 11 | 7043 |
| Insurance-co (IO) [url] | 83 | 0 | 2 | 5822 |

in the face of both missing values and noisy data, thereby enhancing interpretability and demonstrating their robustness.

**FT-Transformer** [Gorishniy et al., 2021] [9] An adaptation involves the incorporation of the Transformer architecture specifically tailored for tabular data processing. A distinct Feature Tokenizer is employed to transform individual columns into continuous vector spaces. Numerical values are subjected to multiplication with a training vector, while other non-numeric data is tokenized as text, utilizing word embedding techniques for further processing.

**GANDALF** [Joseph and Raj, 2023] [10] A deep learning framework, denoted as the Gated Adaptive Network for Deep Automated Learning of Features (GANDALF), which demonstrates its promising performance, interpretability, and computational efficiency in the context of tabular data analysis. GANDALF incorporates a special tabular processing unit characterized by a gating mechanism and integrated feature selection capabilities, termed the Gated Feature Learning Unit (GFLU), serving as a fundamental component for feature representation learning.

**NODE** [Popov et al., 2019] [11] The Neural Oblivious Decision Ensembles (NODE) constitute an emerging deep learning architecture expressly engineered for heterogeneous tabular data. The NODE architecture extends the concept of ensembles composed of oblivious decision trees, harnessing the advantages inherent in both end-to-end gradient-based optimization and the potent capabilities of multi-layer hierarchical representation learning.

**TabNet** [Arik and Pfister, 2021] [12] TabNet represents an interpretable canonical deep learning architecture designed specifically for the represention of tabular data. Its core mechanism employs sequential attention to systematically determine the pertinent features to consider at each decision-making juncture. This strategic feature selection not only enhances interpretability but also optimizes the learning process by allocating learning capacity to the most salient features.

**AutoInt** [Song et al., 2019] [13] AutoInt is an automated mechanism designed for the autonomous acquisition of high-order feature interactions inherent in input attributes. It performs a joint mapping of numerical and textual features onto a shared, reduced-dimensional space. Subsequently, a multihead self-attentive

---

[9] https://github.com/yandex-research/tabular-dl-revisiting-models
[10] https://github.com/manujosephv/pytorch_tabular
[11] https://github.com/Qwicen/node
[12] https://github.com/sourabhdattawad/TabNet
[13] https://github.com/DeepGraphLearning/RecommenderSystems

Table 8: Ablation results on public tabular datasets (CG ∼ IO).

| Method/Dataset | CG | CA | DS | AD | CB | BL | IO | Avg. |
|---|---|---|---|---|---|---|---|---|
| UniTabE finetune | **0.79** | **0.94** | **0.66** | **0.91** | **0.88** | **0.84** | **0.76** | **0.83** |
| w/o Fuse Layer | 0.74 | 0.84 | 0.60 | 0.90 | 0.80 | 0.83 | 0.69 | 0.77 |
| w/o Linking Layer | 0.69 | 0.84 | 0.61 | 0.90 | 0.72 | 0.81 | 0.69 | 0.75 |
| w/o Fuse&Linking | 0.65 | 0.85 | 0.61 | 0.84 | 0.65 | 0.77 | 0.65 | 0.72 |
| w/o MCM | 0.77 | 0.92 | 0.58 | 0.91 | 0.88 | 0.84 | 0.75 | 0.81 |
| w/o CL | 0.74 | 0.84 | 0.60 | 0.90 | 0.90 | 0.83 | 0.69 | 0.79 |
| MCM rate: 0.3 | 0.79 | 0.92 | 0.65 | 0.90 | 0.84 | 0.83 | 0.75 | 0.81 |
| MCM rate: 0.2 | 0.78 | 0.92 | 0.64 | 0.90 | 0.82 | 0.85 | 0.73 | 0.81 |
| MCM rate: 0.1 | 0.79 | 0.93 | 0.65 | 0.91 | 0.85 | 0.83 | 0.75 | 0.82 |
| 3L Decoder | 0.75 | 0.93 | 0.63 | 0.90 | 0.86 | 0.83 | 0.76 | 0.81 |
| 6L Decoder | 0.74 | 0.90 | 0.62 | 0.89 | 0.84 | 0.83 | 0.73 | 0.79 |
| 12L Decoder | 0.74 | 0.89 | 0.60 | 0.90 | 0.82 | 0.81 | 0.69 | 0.78 |
| 1L Transformer Decoder | 0.78 | 0.93 | 0.62 | 0.91 | 0.87 | 0.84 | 0.73 | 0.81 |
| 3L Transformer Decoder | 0.73 | 0.88 | 0.59 | 0.86 | 0.83 | 0.83 | 0.72 | 0.78 |
| 3L Transformer Decoder | 0.72 | 0.86 | 0.68 | 0.85 | 0.77 | 0.82 | 0.69 | 0.77 |
| 12L Transformer Decoder | 0.74 | 0.86 | 0.60 | 0.89 | 0.79 | 0.81 | 0.69 | 0.77 |

neural network, featuring residual connections, is employed to explicitly capture feature interactions within this low-dimensional representation. Through incorporating multiple layers of the multi-head self-attentive neural network, the model accommodates the modeling of diverse orders of feature combinations derived from the input attributes.

## A.2 DATASETS

The statistical characteristics of each dataset involved in the downstream tasks are provided in Table 7. The table includes the dataset names, along with their corresponding abbreviations, in the "**Name**" column. Additionally, we have included hyperlinks to access each dataset directly within the table for reference. We have deliberately selected these datasets, representing diverse domains (e.g. finance, medicine, healthcare, real estate, credit analysis, etc.), with the primary objective of conducting a comprehensive and thorough evaluation of our methodology.

## A.3 EXPERIMENTAL DETAILS

The UniTabE is pretrained with ten steps of gradient accumulation. The two pretraining objectives, multi-cell-masking and contrastive learning, are carried out interactively. We adopt the hyperparameter $p$ to control the overall proportion of cells to mask. The program generates random number $q \in [0, 1]$ for each cell. If $q \leq p$, then the corresponding cell will be masked. Masking single cell is used as the backup option in some cases, such as there is no any cell to mask. $p$ is set to be 0.15 in this work. We follow prior work [Wang and Sun, 2022], the overlap ratio used in this work is set to be 50% while conducting contrastive learning. We skip those tables whose column numbers are less than two during training with contrastive learning objective. Furthermore, we employ a random shuffling procedure to alter the arrangement of columns within the table during the pretraining phase. The batch size is set as 8 while finetuning our model in downstream tasks. The finetune learning rate is 1e-6. We use the same optimizer settings as that used during pretraining. For compared baselines, we use the official optimal settings and train these models on different datasets. We then average scores of five runs for all model for fair comparison.

## A.4 ABLATION STUDIES

Table 8 presents the detailed ablation results on different datasets.

## A.5 TABLEQA CASE STUDY

We have observed a diminishing performance trend in classification tasks as we increase the number of decoder layers in our model. This phenomenon raises the suspicion that a strong decoder may inadvertently

Table 9: Comparison of text generation ability. The BLEU score between ground truth and generated text. The results reveal that more pretrained knowledge are stored in decoder as its layer number grows.

| # Layers/Decoder | LSTM | Transformer |
|---|---|---|
| 1L | 24.72 | 24.91 |
| 3L | 27.00 | 27.42 |
| 6L | 28.82 | 29.14 |
| 12L | 29.54 | 30.16 |

Table 10: Performance comparison in filling in missing value. "mean/mode" uses the average in numerical column as prediction, and takes the most common text as prediction for textual column.

| Method/Dataset | Regression (MAE ↓) | | | | | | Text Generation (BLEU ↑) | | | | | |
|---|---|---|---|---|---|---|---|---|---|---|---|---|
| | CG | CA | DS | AD | CB | BL | CG | CA | DS | AD | CB | BL |
| mean/mode | 0.81 | 0.61 | 0.84 | 0.78 | 0.73 | 0.84 | 48 | 62 | 46 | 62 | 65 | 43 |
| Linear Regress | 0.64 | 0.58 | 0.91 | 0.45 | 0.60 | 0.51 | - | - | - | - | - | - |
| GPT-3 | 0.58 | 0.57 | 0.69 | 0.64 | 0.61 | 0.46 | 48 | 66 | 39 | 73 | 71 | 82 |
| UniTabE scratch | 0.49 | 0.56 | 0.83 | 0.69 | 0.72 | 0.38 | 35 | 63 | 28 | 67 | 18 | 75 |
| UniTabE finetune | **0.40** | **0.51** | **0.61** | **0.43** | **0.51** | **0.22** | **59** | **76** | **51** | **80** | **85** | **92** |

divert some of the pretrained knowledge away from the encoder, potentially compromising the encoder's capacity for representation. In contrast, a weaker decoder places the onus of modeling knowledge on the encoder during pretraining. Consequently, an encoder trained in collaboration with shallow decoder tends to yield better semantic representations. To assess the efficacy of text generation, we constructed a table question answering (table QA) dataset based on the CG dataset. In this dataset, we omitted the "purpose" column from the original data and employed the prompt "why the person applies for the loan?" for finetuning our UniTabE model. Table A.5 presents a comparative analysis of results across varying decoder sizes. Notably, we observe that the BLEU score exhibits an ascending trend as the number of decoder layers increases. Given the observation that pretrained models equipped with a powerful decoder demonstrate marginally inferior performance in semantic representation, we have opted for the design featuring a shallow decoder, aiming to prioritize the storage of acquired knowledge within the encoder, as opposed to relying on a powerful decoder or applying the balanced encoder-decoder architecture. This phenomenon aligns with findings in prior research [Kasai et al., 2020; Sun et al., 2021; Li et al., 2021; Kong et al., 2022] which further inspired us to take such architecture. For instance, all of them adopted Transformer as backbone with shallow decoder. Extensive experiments of Kasai et al. [2020] and Kong et al. [2022] demonstrated competitive performance between the single-layer decoder integrating with strong encoder and the balanced encoder-decoder depth in Neural Machine Translation (NMT). In addition, the LSTM decoder underperforms the Transformer decoder in the generation of target text, underscoring the suitability of the LSTM decoder for deployment as a weaker component within our model.

## A.6 FILL IN MISSING VALUE ANALYSIS

We evaluate the model's capability to fill in missing values, utilizing the Mean Absolute Error (MAE) metric for numerical columns and the BLEU score for textual predictions. For comparative analysis, we also fine-tune and test the GPT-3 model. The results, outlined in Table 10, indicate a significant improvement over other baseline models. This substantial gain in performance further corroborates the efficacy of our pretrained model in handling missing values, demonstrating its potential for applications in data completion and recovery tasks.

**Input:**

| psvolfilm | pcut | pcut | pcut | Timestamp | pcut | psvolfilm | psvolfilm |
|-----------|------|------|------|-----------|------|-----------|-----------|
| 0.88085 | -0.02716 | -0.46049 | 906.37207 | 5.924 | 136122816.0 | 288624243.0 | 5521.72217 |
| 1.10703 | -0.00099 | 0.55356 | -4164.50488 | [MASK] | 136124253.0 | 288622077.0 | 4150.10059 |
| 0.97426 | -0.04228 | 0.62234 | -2770.6145 | 5.428 | 136124555.0 | 288621753.0 | 3317.0603 |
| 0.92573 | 0.04246 | -0.57147 | 549.31641 | 1.912 | [MASK] | 288607277.0 | 5569.54053 |
| -0.45747 | 136123573.0 | 0.01218 | 5521.72217 | 288625307.0 | 6749.7251 | 0.79271 | 6.116 |
| 4116.43945 | 0.648 | 136116864.0 | 0.01266 | 288602476.0 | 3050.28613 | -0.08359 | 0.75895 |
| … | … | … | … | … | … | … | … |

**Prompt:** fill in the missing value, Timestamp:

**Target**: 5 . 5 1 6

Figure 5: Demonstration of multi-cell-masking.

### A.7 DEMONSTRATION OF MULTI-CELL-MASKING

Figure 5 illustrates the pretraining example of multi-cell-masking. We use the prompt template "fill in missing value, <column name> :" to specify the precise masked column to predict.

### A.8 KAGGLE DATA LICENSE AND COPYRIGHT

Our utilization of Kaggle's dataset for UniTabE pretraining adheres meticulously to Kaggle's terms of use and licensing agreements. The publicly available dataset complies with Kaggle's licensing terms, and any specific requirements imposed by dataset providers have been respected. The dataset was collected through the official Kaggle API (https://github.com/Kaggle/kaggle-api), specifying the license option as "gpl" and "odb". We exclusively collected data under licenses such as GNU General Public License (GPL) and Open Data Commons series (e.g., ODbL, PDDL, ODC-By), ensuring compatibility for open-sourcing our model. For example, the "sales-from-different-stores" dataset is obtained under the ODbL license (https://opendatacommons.org/licenses/odbl/1-0/), allowing a worldwide, royalty-free, non-exclusive license for data use. Our approach aligns with copyright laws, and we uphold ethical considerations, prioritizing responsible and fair data use within permitted scope and respecting Kaggle's data privacy policies. We intend to include information on data licensing, copyright, and ethical considerations in the final version of our paper if it is accepted.

