# OpenReview forum: "UniTabE: A Universal Pretraining Protocol for Tabular Foundation  Model in Data Science"
_ICLR.cc/2024/Conference — ICLR 2024 poster_

### Official Review · Reviewer_KBNh · 2023-10-23

**Soundness:** 4 excellent
**Presentation:** 3 good
**Contribution:** 3 good
**Rating:** 6
**Confidence:** 5

**Summary:**

The paper delves into the application of large language models (LLM) to tabular data, presenting a novel approach utilizing the transformer model.

To address the unique characteristics of tabular data, the authors introduce "TabUnit", a method to amalgamate data type, column name, and cell values into comprehensive cell vectors. These vectors are then processed through transformer-based encoders and fed into a shallow LSTM decoder to regenerate tokens for the target label column.

Upon pre-training the UniTabE model using 13B samples from Kaggle, the authors fine-tuned it on 12 chosen classification and evaluation datasets. The results reveal a significant improvement over prior methods that lack pre-training.

**Strengths:**

1. Quality:
The paper's quality stands out as it presents a model pre-trained on an impressively large tabular prediction dataset encompassing 13B samples. The authors conducted a meticulous evaluation of the model with contemporary tabular prediction baselines.

2. Significance:
The results offered by the authors are particularly enlightening. While they tout the enhanced performance derived from pre-training on expansive tabular data, a closer examination (as reflected in Table 3) indicates only a slight uplift in downstream task performance post-fine-tuning, especially given the model's exposure to 13B samples. This subtle observation underscores the intricate challenges posed by tabular data prediction, taking into account its inherent heterogeneity and noise. It's pivotal to recognize that these findings could potentially delineate a performance benchmark for similar BERT-inspired methods in tabular prediction. This could galvanize researchers to delve deeper into innovative tabular modeling paradigms and re-evaluate data processing strategies for tabular prediction.

**Weaknesses:**

1. Evaluation:
Given the vastness of the training data, with 13B samples spanning 303 diverse domains, the evaluation datasets chosen seem somewhat narrow in scope. The comprehensive variety of training domains calls for an equivalently diverse evaluation to genuinely test the model's versatility and adaptability. Authors should consider widening their evaluation scope by leveraging an even more diverse array of tabular datasets.

2. Results:
While Table 3 does indicate that UniTabE marginally outperforms other baselines, the incremental gain appears to be disproportionate to the extensive computational overhead of the pre-training phase. It's noteworthy that the performance between a ground-up UniTabE (from scratch) and a pre-trained plus fine-tuned UniTabE are remarkably similar, prompting questions about the efficiency of the elaborate pre-training process.

3. Method:
The paper's tab unit processing technique, although novel in its implementation, seems to draw heavily from precedents like TransTab and FT-Transformer. Additionally, the supervised strategies, such as multi-cell masking and contrastive learning, bear significant resemblance to methodologies detailed in prior research.

**Questions:**

1. Table 3 Variants:
The descriptions of the three UniTabE variants presented in Table 3—namely scratch, +XGBOOST, and fine-tune—appear to be absent. Notably, UniTabE scratch seems to rival the performance of UniTabE fine-tune and even surpasses UniTabE+XGBOOST in many instances. Given that UniTabE has been exposed to 13B tabular samples, this outcome raises questions. What factors might account for this similarity in performance, especially between the scratch and fine-tuned versions? Moreover, would broadening the evaluation across a more diverse set of datasets provide deeper insights into the tangible benefits of UniTabE pre-training?

2. Llama2 13B Training:
Regarding Llama2's tabular prediction training, how exactly was the model trained to handle tabular data predictions?

---

> ### Author Response · Authors · 2023-11-18
>
> Thank you sincerely for your reviewing and insightful suggestions.
>
>
> **Diversity of downstream datasets:**
>
> The selection of downstream evaluation datasets was a deliberate and considered process. Despite the expansive nature of our training data, covering 303 domains with 13 billion samples, ensuring a representative evaluation across all domains is challenging. We intentionally opted for specific downstream datasets in domains(e.g. investing, finance, healthcare, real estate, etc). These domains not only align with our pretraining data but also represent a diverse range of practical applications. The diversity of the downstream datasets is discussed in detail in Appendix A.2 (DATASETS). We acknowledge the importance of a comprehensive evaluation and would like to explore additional diverse tabular datasets to further enhance the breadth of our evaluation scope.
>
>
> **The pretraining methodology:**
>
> Our method is clearly different from TransTab and FT-Transformer.
>
> TransTab employs contrastive learning on a limited set of tables from the same clinical domain, lacking the large-scale pretraining on diverse domains and the objective of multi-cell-masking undertaken in our work. Furthermore, TabUnit's cell processing strategy differs significantly from that of TransTab. Our model processes each cell in the model individually, while TransTab divides tabular data into three types and adopts textualization to convert each type of data into text before feeding it into an embedding module.
>
> FT-Transformer, despite sharing a similar cell-wise processing concept, differs in representation and linking layer mechanisms from our approach. Our method involves utilizing a sequence of vectors for each cell, encompassing information on both the column name and cell value, with a specialized linking layer maintaining their relationship. We convert numerical values into a sequence of digits to preserve the original recording structure and numerical significance, while non-numerical values are treated as text and passed into the embedding module. In contrast, FT-Transformer considers only the cell value and represents each cell with a single vector. For numerical values, FT-Transformer acquires cell representations by multiplying them with a learning vector, while non-numerical values are mapped into a singular vector through the embedding module. Importantly, FT-Transformer is not designed for pretraining purposes.
>
>
> **The pretraining objective:**
>
> We think that applying the general idea of mask-then-predict is not a weakness of this work. The mask-then-predict idea is a simple but effective approach of providing self-supervised optimization, and is widely used in pretraining, especially for the pretraining in NLP. The application of the mask-then-predict paradigm in this work is not perceived as a weakness but rather as a deliberate choice. Our multi-cell-masking objective is the crafted variance of mask-then-predict. We manipulate the masking at the cell level with the consideration that filling in the entire content of a cell is more useful than merely filling in part of the cell content in practical applications. In addition, this objective can be regarded as learning the overall similarity of the table, while the contrastive learning focuses on learning the inner differences within the same table. This would bring the benefit of obtaining hierarchical understanding of the tabular data, distinguishing it from existing methods.
>
>
> **To Q1:**
>
> "UniTabE scratch" is a model trained on the downstream dataset without utilizing pretrained parameters. "UniTabE finetune" represents the model finetuned from pretrained parameters. "UniTabE + XGBoost" denotes the XGBoost model incorporating both the original input and the tabular representation generated by UniTabE, as discussed in the "Learned Feature + XGBoost" paragraph of our paper. In Tables 2 and 3, "UniTabE finetune" exhibits promising improvement compared to "UniTabE scratch." The results in Table 4 underscore the significant performance enhancement achieved through pretraining in the context of zero-shot prediction. Notably, compared to XGBoost, "UniTabE + XGBoost" demonstrates improved performance across the majority of evaluated datasets, indicating that the learned tabular representation from UniTabE contributes positively to XGBoost, even if the improvement on certain datasets is modest.
>
>
> **To Q2:**
>
> In the training process for Llama2's tabular predictions, we followed a specific template for both finetuning and predicting. Each example was structured as follows:
>
> "I will provide you with a table. Your task is to predict the classification result based on this table, with options being 0 or 1. <table-description> The subsequent text contains the content of the table in Markdown: <table-content-in-markdown>."
>
> We use the Hugging Face's transformers module as the framework, in which "LlamaForSequenceClassification" is used, for both finetuning and inferencing Llama2.

---

> ### Author Response · Authors · 2023-11-23
>
> Thank you for dedicating your time to provide valuable feedback. As the deadline for author-reviewer discussions ends today, we kindly invite you to continue the dialogue if you have any further comments.

---

### Official Review · Reviewer_es5d · 2023-10-31

**Soundness:** 3 good
**Presentation:** 3 good
**Contribution:** 3 good
**Rating:** 8
**Confidence:** 3

**Summary:**

This paper presents UniTabE - a universal pretraining method for tabular data that consists of three components: tabular unit decomposition, encoding, and decoding. Compared to prior work, a key novelty of this model is that it is trained from scratch, and it involves innovations in terms of developing a TabUnit splitter and a simple decoder. The compilation of a large dataset for model training is also an interesting undertaking. The evaluation section provides a large number of experiments, showing the improvement brought by UniTabE on fine-tuned and zero-shot classification, regression tasks, and missing data prediction, whereas a series of ablations show the benefit of each aspect of the method.

**Strengths:**

S1) The paper is well-written and easy to follow. It provides information that should suffice to reproduce the presented method.

S2) The methodological contributions are clearly indicated and well-justified by the paper.

S3) The paper contributes a novel dataset and set of models, that should be interesting to practitioners working with tables.

S4) The experimental section shows the benefit of this method against competitive baselines. Ablation studies are provided to ensure that the improvement comes from the expected method components.

S5) To my knowledge, the undertaking to create such a foundational model for tables is novel. I agree with the authors that it can be very impactful given the amount of tabular data online.

**Weaknesses:**

W1) The paper lacks a discussion of the ethical and legal aspects of the data, including biases, copyright, and licensing.

W2) The proposed method has a strong performance against the baselines consistently, but the baseline choices are inconsistent across the experiments. Can the authors motivate the reason for selecting certain baselines for each experiment, especially for Tables 4 and 5?

W3) The paper talks about generalization to tables with more columns, but the generalization to tables with more complicated structures (e.g., multi-row and multi-column blocks) is not discussed. I am wondering how are such tables handled by the UniTabE.

Minor:
* Figure 3a is impossible to read - please enlarge the figure for the follow-up version, and ideally provide more guidance/commentary about it in the paper.
* Typo: "Expect for " -> "Except for"
* The commentary of Table 3 (page 9) is unclear, as it refers to "a slight performance enhancement", even though the two versions seem to perform more-or-less on par.

**Questions:**

Q1) Given that a pretty large amount of data from Kaggle is used for pretraining of UniTabE, I would appreciate it if the authors could comment on the aspect of data licensing, copyright, and ethics of use. Also, what would the use of this model as a foundational model for tables entail given the specific tables used for pretraining?

Q2) Can the authors motivate the reason for selecting certain baselines for each experiment, especially for tables 4 and 5?

Q3) How does UniTabE handle tables with more complicated structures, where the organization of columns and rows is less obvious?

**Details Of Ethics Concerns:**

Given that the authors use a large amount of data from Kaggle, I was surprised to not see any discussion on the licensing, ethics of use, and copyright of this data. I would appreciate it if the authors could clarify this aspect.

---

> ### Author Response · Authors · 2023-11-18
>
> Thank you very much for your thorough review, insightful suggestions, and for pointing out the typos!
>
> **To Q1:**
>
> ***Data licensing, copyright, and ethics:***
>
> Our utilization of Kaggle's dataset for UniTabE pretraining adheres meticulously to Kaggle's terms of use and licensing agreements. The publicly available dataset complies with Kaggle's licensing terms, and any specific requirements imposed by dataset providers have been respected. The dataset was collected through the official Kaggle API (https://github.com/Kaggle/kaggle-api), specifying the license option as "gpl" and "odb". We exclusively collected data under licenses such as GNU General Public License (GPL) and Open Data Commons series (e.g., ODbL, PDDL, ODC-By), ensuring compatibility for open-sourcing our model. For example, the "sales-from-different-stores" dataset is obtained under the ODbL license (https://opendatacommons.org/licenses/odbl/1-0/), allowing a worldwide, royalty-free, non-exclusive license for data use. Our approach aligns with copyright laws, and we uphold ethical considerations, prioritizing responsible and fair data use within permitted scope and respecting Kaggle's data privacy policies. We intend to include information on data licensing, copyright, and ethical considerations in the final version of our paper if it is accepted.
>
>
> ***Use of our model as a foundamental model:***
>
> 1) Understanding tabular data: UniTabE serves as a powerful foundational model for understanding tabular data. It facilitates intuitive obtaintion of tabular representation through natural language prompts, enhancing usability in downstream applications.
>
> 2) Tabular data synthesis: UniTabE, trained to predict missing values in tables, holds the potential for synthesizing tabular data. By masking entire rows as missing values, the model can generate synthesized data, contributing to privacy protection by excluding real, sensitive information. Synthesized data further benefits applications such as data augmentation, enhancing diversity, and simulating scenarios.
>
> 3) Combining with large language models (LLMs): Inspired by the success of diffusion models in text-to-image tasks, future research may explore utilizing our pretrained model as a foundational component to generate semantic representations for tabular data combining with LLMs within the diffusion framework.
>
>
> **To Q2:**
>
> 1) Baselines selection for Table 2 and Table 3:
> The 12 datasets(PID\~HPA) in Table 2 (in our paper) come from Kaggle. Despite their exclusion from our pretraining data, we just conducted a preliminary comparison of our approach with TransTab-LSTM (a representative Transformer-based model), XGBoost, and "UniTabE scratch" on these datasets, considering potential domain similarities to pretraining data domains. Additionally, we employed other public datasets (CG\~IO) for comprehensive evaluation in Table 3, ensuring a fair comparison with prior methodologies that may not have been trained on Kaggle's data domains.
>
>
> 2) Baselines selection for Table 4 and Table 5:
> In Table 4, we concentrated on evaluating zero-shot prediction performance. Given the limitations of most existing models that cannot be applied to table structures they were not trained with, hindering their capability for zero-shot prediction, we exclusively compared our model against a finetuned and a randomly initialized model.
> In Table 5, our objective was to assess the scalability of models in handling incremental columns added to tables(e.g. in clinical trials, incremental columns are collected across different phases). Similar to Table 4, we excluded methods that cannot support table structures different from those encountered during training; for example, TabPFN prohibits a different table structure at the test stage compared to the trained one. We chose TransTab-LSTM as a baseline for its adaptability to incremental columns and its representation as a Transformer-based model. Additionally, we incorporated "UniTabE scratch" as a comparative model to evaluate the benefits derived from pretraining.
>
>
> **To Q3:**
>
> We employ TabUnit to process each cell, transforming the column name and value into a sequence of hidden states that serve as the cell's representation. For tables with multi-tiered headers, we adopt the format "top-level-header-name :: second-level-header-name :: ..." to convey the nested structure. The following table presents multi-tiered headers, with actual data omitted for clarity of demonstration:
>
> | Models | &nbsp;&nbsp;&nbsp;&nbsp; Task 1 | | &nbsp;&nbsp;&nbsp;&nbsp; Task 2 | |
> |----|:----:|:----:|----:|:----:|
> |        | **Precision** | **Recall** | **Precision** | **Recall** |
> | c11 | c12 | c13 | c14 | c15 |
> | c21 | c22 | c23 | c24 | c25 |
>
> The column name for cell c11 is "Models", while the column name for cell c12 and c22 is set as "Task 1 :: Precision". Utilizing our feature processing module, TabUnit, our trained model exhibits flexibility in encoding tables with complex structures.

---

> ### Author Response · Authors · 2023-11-23
>
> Thank you for dedicating your time to provide valuable feedback. As the deadline for author-reviewer discussions ends today, we kindly invite you to continue the dialogue if you have any further comments.

---

### Official Review · Reviewer_3wag · 2023-11-01

**Soundness:** 2 fair
**Presentation:** 3 good
**Contribution:** 2 fair
**Rating:** 5
**Confidence:** 3

**Summary:**

The paper proposes a transformer model for table classification and regression. The authors curated a large set of Kaggle datasets for training and evaluation. The central component of the proposed architecture is the TabUnit, which embeds data type, column name and column value respectively, and fuses as well as links the embeddings via gates. The authors propose to use a Transformer encoder and LSTM as decoder. The model is pre-trained by different strategies, namely cell and column masking and siamese triplet loss based on (partial-) row contents. Fine-tuning then entails predicting missing columns for a new dataset specialisied prompts for tasks such as question answering. The authors trained the Transformer on a self-collected dataset comprising 13 billion tables. The approach is evaluated for standard, fine-tuned as well as combined version (with XGBoost). Part of the evaluation is done for Kaggel tasks (12), where the approach is compared to XGBoost and TransTab. The rest of the evaluation is performed for 7 public datasets, where a larger amount of competitors is compared against. The results show that the proposed approach often achieves superior performance in terms of AUC/R2. In addition, multiple ablation studies also highlight, among others, robustness of the approach towards missing data.

**Strengths:**

The chosen topic of the paper is highly relevant to the venue, as tabular machine learning is, as argued in the paper, often overlooked but highly important in industrial applications.

The proposed approach to learn designated embeddings for data type, column name and values within the TabUnit is sensible and is different to existing approaches. In addition, the authors collect a large dataset for training their Transformer. It would be value adding to open-source such a model.

The work incorporates a large number of competitors in part of the evaluation (on open datasets): The authors include more than 10 baseline approaches of recent papers to compare the performance of their approach on a selected sample of open datasets. The chosen baselines, here, are well-chosen. The authors also provide ablation studies wrt to ther model design, highlighting the implications of omitting parts of the architecture. Lastly, the predictive performance of the approach is often better than the competitors.

**Weaknesses:**

It is unclear why only part of the evaluation was done for all Transformer-based competitors and why no regression results for a XGBoost regressor have been generated. It would have been highly insightful to have the same evaluation as for the open datasets.

It would also have been insightful to compare to a representative AutoML approach in order to see the boundaries of the approach. Of course, it does not have to be assumed that the model has to beat the best tuned models, but it should come close enough.

While the results are promising, the gap between the competitors is often rather small. This is not to reduce the quality of the predictive performance, as the task at hand is very difficult and competitive, it just shows that competitors are already strong.

While part of the evaluation is exhaustive in terms of baseline approaches, the evaluation contains relatively little open datasets compared to some of the competing papers. It would be beneficial to add more thorough design decisions to why the datasets have been chosen or why they are sufficiently representative.

The ablation study with respect to robustness is very interesting and highlights another benefit of the method - being able to deal with missing data. It would be intersting to know if this result holds over the other competitors too.

A minor point (as many competitors were looked at): It would have intersting to see the performance of TUTA, as similar to the proposed approach, more refined embeddings are learned.

**Questions:**

Will get code / the trained model be made available?

Why was the evaluation only in part conducted with all competitors?

What are the most challenging datasets you tested the method on?

What are other advantages of the proposed approach next to pure predictive performance? Is the method's robustness superior to the other competitors as well?

---

> ### Author Response · Authors · 2023-11-18
>
> Thank you very much for your reviewing and insightful suggestions!
>
>
> **Diversity of Downstream Datasets:**
>
> Our choice of downstream datasets was deliberate, encompassing diverse domains, including investing, finance, medicine, healthcare, real estate, dress sales, and so on. The selection aimed at providing a comprehensive evaluation across practical domains, as detailed in Table 7.
>
>
>
> **Performance gap between the competitors:**
>
> While competitors are strong, our method consistently outperforms them on most datasets. We acknowledge the modest improvements in certain tasks (e.g., RSP, AD). However, the average score on CG\~IO datasets demonstrates a clear improvement against prior approaches.
>
>
>
> **Ability of dealing with missing values:**
>
> In practical applications, modeling tabular data with existing missing values is common. Our proposed model is inherently equipped to effectively address missing values, as it is trained with a mask-the-predict objective. In Table 10, we conducted an evaluation of our model and competitors (including GPT) for filling in both numerical and textual missing values. It's noteworthy that a considerable number of prior methodologies do not accommodate this scenario, especially in predicting textual values, unless explicitly trained for such cases. For example, TabPFN can be additionally trained to predict numerical values, similar to regression tasks, but lacks support for filling in textual values. In contrast, our method can perform this task without requiring further training.
>
>
>
> **To Q1:**
>
> we plan to release our trained model and our code.
>
>
>
> **To Q2:**
>
> We further evaluated the performance of XGBoost in regression tasks and TUTA in classification tasks. Results are as follows.
>
> |method/Dataset|MIP|GPP|RSP|CCL|MMP|HPA|
> |----|----|----|----|----|----|----|
> |XGBoost|0.54|0.64|0.98|0.78|0.71|0.49|
> |UniTabE|0.75|0.99|0.99|0.96|0.87|0.58|
>
> |method/Dataset|CG|CA|DS|AD|CB|BL|IO|Avg.|
> |----|----|----|----|----|----|----|----|----|
> |TUTA|0.74|0.92|0.61|0.81|0.74|0.83|0.65|0.76|
> |UniTabE|0.79|0.94|0.66|0.91|0.88|0.84|0.76|0.83|
>
> The 12 datasets(PID\~HPA) in Table 2 (in our paper) come from Kaggle. Despite their exclusion from our pretraining data, we just conducted a preliminary comparison of our approach with TransTab-LSTM (a representative Transformer-based model), XGBoost, and "UniTabE scratch" on these datasets, considering potential domain similarities to pretraining data domains. Additionally, we employed other public datasets (CG\~IO) for comprehensive evaluation, ensuring a fair comparison with prior methodologies that may not have been trained on Kaggle's data domains.
>
>
>
> **To Q3:**
>
> Most methods faced considerable difficulties in handling datasets with pronounced class imbalances, exemplified by the highly skewed class distribution in the IO dataset (around 1:16 for positive label:negative label). Predicting minority classes accurately is challenging in such imbalanced settings. Our method exhibits a distinct relative improvement of about 7% compared to Transformer-based methods.
>
>
>
> **To Q4:**
>
> Except for the predictive performance, our method has other advantages:
>
> 1) Adaptability: Our model exhibits adaptability to diverse table schemata, leveraging TabUnit for versatile processing. The representation of each cell, akin to tokens in natural language processing, allows the application of Transformer-based architecture talented in sequence modeling. This inherent flexibility enables our model to effectively encode representations of cells, making it adaptable to various tables, analogous to pretrained language models handling different texts.
>
> 2) Flexibility: The proposed approach is designed for versatility, adapting seamlessly to different tasks within a unified framework. Equipped with a decoder, our model operates in a table-to-sequence format, encompassing classification, regression, and text generation. This flexibility enhances its applicability across a spectrum of scenarios.
>
> 3) Robustness: Evaluation on CG\~IO datasets outside Kaggle's domains demonstrates our method's continued superior performance. Zero-shot classification experiments (Table 4) further affirm its robustness in out-of-domain scenarios. Notably, our model exhibits robustness in the presence of missing values in tabular data, as demonstrated in Table 10.
> Moreover, the AUC scores in Table 3 are computed as the average scores across 5 training runs. The mean standard deviation derived from five iterations on all datasets is presented bellow. UniTabE exhibits a lower variance compared to other competitive methods, indicating its consistency and reliability across different runs.
>
> |Method|Avg. Standard Deviation|
> |----|----|
> |UniTabE|0.017|
> |XGBoost|0.023|
> |Tapas|0.024|
> |TabPFN|0.025|
> |TaBERT|0.041|
> |TabTransformer|0.027|
> |FT_Transformer|0.024|
> |GANDALF|0.021|
> |NODE|0.032|
> |TabNet|0.045|
> |AutoInt|0.029|
> |TUTA|0.021|

---

> ### Author Response · Authors · 2023-11-23
>
> Thank you for dedicating your time to provide valuable feedback. As the deadline for author-reviewer discussions ends today, we kindly invite you to continue the dialogue if you have any further comments.

---

### Public Comment · ~Quan_Gan1 · 2024-04-14
**Which part of the model do you fine-tune?**

Thanks for your paper.  Very inspiring!

During fine-tuning, may I know whether you adjust the weights of the entire model, or just a part of it?  If the latter, which part do you adjust?

---

### Public Comment · ~Milad_Abdollahzadeh1 · 2025-02-12
**Clarification on the Embedding Function Architecture in TabUnit**

Thank you for sharing this interesting paper—I really enjoyed reading it.

$ $

I have a question regarding the details of the architecture. In the paper, you mention that TabUnit employs an embedding function for column values ($Emb$ in Eq. (3)), which is also used for encoding the prompts for the decoder. However, I couldn’t find details about the architecture of this embedding function. Is it a pre-trained model, or is it trained during pre-training?

I’m particularly interested in how this function handles numerical, textual, and categorical data. As you noted, textualizing numerical data and feeding it into LLMs is not an optimal approach for processing tabular data. Understanding your method’s approach to this challenge would be valuable.

$ $

Thanks in advance for your time!

---

### Meta-Review · Area_Chair_Cxws · 2023-12-08

**Metareview:**

This study introduces UniTabE, a method for universal pretraining of tables in data science, addressing challenges posed by diverse table structures. UniTabE employs TabUnit modules and Transformer encoders to process tables uniformly, allowing for generalizability across tasks and adaptability to various applications. The model's pretraining phase utilizes a curated tabular dataset of approximately 13 billion samples from Kaggle. Rigorous experiments demonstrate UniTabE's superior performance in classification and regression tasks, highlighting its potential to enhance the semantic representation of tabular data and make significant strides in tabular data analysis.

This paper underwent review by three evaluators, with two recommending acceptance and one suggesting a borderline reject. All reviewers agreed on the paper's novelty and significance in proposing pretraining from a substantial amount of tabular data, finding the content easy to follow. Concerns about variations in baselines across experiments were raised, but the authors adequately addressed them in the author response. Despite a somewhat marginal performance improvement, I believe these results still offer valuable insights to the ML community and encourage continued research in this domain. To avoid potential skepticism by readers regarding inconsistent baselines, it is recommended that the final version incorporates additional experimental results during the rebuttal and include through explanations on them (particularly in cases with some constraints like those seen in tables 4 and 5).

**Justification For Why Not Higher Score:**

while the trial by the author is very relevant and meaningful, unfortunately experimental results are not that surprising.

**Justification For Why Not Lower Score:**

the paper has good enough contribution to publish.

---

### Decision · Program_Chairs · 2024-01-16

Accept (poster)